# Thermal Stability Enhancement of L-Asparaginase from *Corynebacterium glutamicum* Based on a Semi-Rational Design and Its Effect on Acrylamide Mitigation Capacity in Biscuits

**DOI:** 10.3390/foods12234364

**Published:** 2023-12-03

**Authors:** Huibing Chi, Qingwei Jiang, Yiqian Feng, Guizheng Zhang, Yilian Wang, Ping Zhu, Zhaoxin Lu, Fengxia Lu

**Affiliations:** College of Food Science and Technology, Nanjing Agricultural University, Nanjing 210095, China; t2023091@njau.edu.cn (H.C.); 2021121002@stu.njau.edu.cn (Q.J.); 2022808095@stu.njau.edu.cn (Y.F.); 2022108054@stu.njau.edu.cn (G.Z.); 2021808098@stu.njau.edu.cn (Y.W.); pingzhu@njau.edu.cn (P.Z.); fmb@njau.edu.cn (Z.L.)

**Keywords:** *Corynebacterium glutamicum*, L-asparaginase, thermal stability, consensus design, acrylamide, biscuits

## Abstract

Acrylamide is present in thermally processed foods, and it possesses toxic and carcinogenic properties. L-asparaginases could effectively regulate the formation of acrylamide at the source. However, current L-asparaginases have drawbacks such as poor thermal stability, low catalytic activity, and poor substrate specificity, thereby restricting their utility in the food industry. To address this issue, this study employed consensus design to predict the crucial residues influencing the thermal stability of *Corynebacterium glutamicum* L-asparaginase (CgASNase). Subsequently, a combination of site-point saturating mutation and combinatorial mutation techniques was applied to generate the double-mutant enzyme L42T/S213N. Remarkably, L42T/S213N displayed significantly enhanced thermal stability without a substantial impact on its enzymatic activity. Notably, its half-life at 40 °C reached an impressive 13.29 ± 0.91 min, surpassing that of CgASNase (3.24 ± 0.23 min). Moreover, the enhanced thermal stability of L42T/S213N can be attributed to an increased positive surface charge and a more symmetrical positive potential, as revealed by three-dimensional structural simulations and structure comparison analyses. To assess the impact of L42T/S213N on acrylamide removal in biscuits, the optimal treatment conditions for acrylamide removal were determined through a combination of one-way and orthogonal tests, with an enzyme dosage of 300 IU/kg flour, an enzyme reaction temperature of 40 °C, and an enzyme reaction time of 30 min. Under these conditions, compared to the control (464.74 ± 6.68 µg/kg), the acrylamide reduction in double-mutant-enzyme-treated biscuits was 85.31%, while the reduction in wild-type-treated biscuits was 68.78%. These results suggest that L42T/S213N is a promising candidate for industrial applications of L-asparaginase.

## 1. Introduction

In modern times, baking and frying have become indispensable elements in the daily lives of humans. However, foods that undergo thermal processing may contain compounds that have detrimental effects on health, such as acrylamide [1]. Acrylamide is primarily formed through the reaction of free amino acids (L-asparagine) with reducing sugars as part of the Maillard reaction [2]. The International Agency for Research on Cancer (IARC) has classified acrylamide as a Group 2A carcinogen, known to be neurotoxic, immunotoxic, and toxic to reproduction [3]. This is of great concern due to its toxicity and widespread presence in thermally processed foods [4]. Various strategies have been used to inhibit the formation of acrylamide, including the careful selection of raw materials, the regulation of processing conditions (temperatures and times), the addition of inhibitors (such as amino acids, organic acids, salts, and glutathione), the addition of plant extracts (phenolics and antioxidants), and the utilization of additives (specific microorganisms and L-asparaginase) [5,6]. Among these approaches, enzymatic treatment is considered to be the simplest and most effective method to inhibit the formation of acrylamide in thermally processed foods without affecting the flavor and nutritional properties of the final product [7].

L-asparaginase is a hydrolytic enzyme that facilitates the hydrolysis of L-asparagine, resulting in the formation of L-aspartic acid and ammonia [8]. The occurrence of acrylamide in food products is primarily attributed to the Maillard reaction between L-asparagine and reducing sugars present in raw materials when subjected to temperatures exceeding 120 °C. However, it is worth noting that the product L-aspartic acid does not participate in the Maillard reaction [9,10]. Consequently, L-asparaginase is predominantly employed to control the formation of acrylamide in food at the source by diminishing the content of L-asparagine, which is a prerequisite substance for acrylamide formation [11]. L-asparaginase is widely distributed in nature, being present not only in microorganisms but also in plants and vertebrates [12]. Among these sources, microorganisms possess several advantages such as easy cultivation, simple isolation and purification, low cost, and easy genetic modification, making them the primary choice for the large-scale production of L-asparaginase [13,14]. However, there is a scarcity of commercially approved microbial-derived L-asparaginases available for use in the food industry. Currently, only two enzymes, Acrylaway^®^ (Novozymes A/S, Bagsvaerd, Denmark), derived from *Aspergillus oryzae*, and PreventASe^TM^ (DSM, Heerlen, Netherlands), derived from *Aspergillus niger* [15,16], have been recognized as “GRAS” (Generally Recognized as Safe) by the United States government. Additionally, the World Health Organization (WHO, Geneva, Switzerland) included them as food additives in 2008 (Series 59) and 2009 (Series 60), respectively [17]. Unfortunately, the existing microbial-derived L-asparaginases suffer from limitations, such as low catalytic activity, poor thermal stability, and inadequate substrate specificity, which hampers their industrial application [18]. Therefore, it is of great importance to molecularly modify the existing microbial-derived L-asparaginase to enhance its industrial properties.

Thermal stability is a crucial characteristic of enzymes as it determines their suitability for industrial applications. Therefore, enhancing the thermal stability of an enzyme is considered a critical and typically necessary step in biosynthesis [19,20]. Enzymes that possess the desired thermal stability exhibit notable advantages in terms of accelerating reactions, increasing evolutionary potential, reducing microbial contamination, and lowering production costs [21,22,23,24]. The catalytic activity of cold-active enzymes at low temperatures is achieved through the flexibility of their structures. However, this increased flexibility also leads to a decrease in the stability of these enzymes. As temperature rises, even reaching room temperature, cold-active enzymes become sensitive to heat, resulting in poor storage stability and making it difficult for them to meet industrial requirements [25,26,27]. Therefore, a thermostable enzyme holds significant and valuable potential for application in the food industry. Currently, there are two main approaches to obtaining L-asparaginase with high thermal stability. The first is to screen for new L-asparaginase genes from extreme environments, while the second involves protein engineering to reconstruct mesophilic L-asparaginase [28,29]. Although several thermophilic L-asparaginases have been extracted from hot springs and hydrothermal vents (such as *Thermococcus* and *Pyrocococcus* sp.), their enzymatic activity and substrate specificity do not meet the requirements for industrial applications [30,31]. Moreover, the screening process for these enzymes is time-consuming and costly [32]. Conversely, these limitations can be overcome through protein engineering (irrational design (directed evolution), semi-rational design, and rational design). Among these, semi-rational design has gained attention from researchers due to its simplicity, short screening period, and high success rate without requiring a comprehensive understanding of the relationship between an enzyme’s structure and function [33]. Consensus design represents a more promising approach to improving enzyme performance as it utilizes small, intelligent, and functional mutation libraries to avoid trade-offs between thermal stability and activity. Consensus design is a sequence-based method that relies on the amino acid sequences of known proteins. It involves comparing the amino acid sequences of target proteins with their family sequences through a multiple-sequence comparison. By calculating the frequency of amino acid occurrences at each site, evolutionarily conserved consensus sequences are obtained. Amino acids that occur with higher frequencies are considered relatively stable throughout evolution [34,35,36]. Conserved amino acid residues within enzyme structures play a crucial role in proper folding, catalytic activity, and stability [37]. This method has been successfully employed to enhance the thermal stability of enzymes [38]. Jiao et al. [39] and Chi et al. [33] utilized this method in conjunction with site-saturated mutagenesis to successfully screen mutant L-asparaginase enzymes with significantly improved thermal stability, demonstrating that the combined approach based on consensus design has potential for modifying enzyme thermal stability.

At present, reducing acrylamide in fried potato products through the pretreatment of raw materials with microbial-derived L-asparaginase is a widely researched field in the food industry. Promising results have been achieved, with acrylamide reductions of up to 90% or more [7,40,41,42]. However, there are a limited number of studies on the application of L-asparaginase in baked foods. *Corynebacterium glutamicum* is extensively used in the food industry as an industrially produced strain with a remarkable level of safety. It is noteworthy that this strain does not generate endotoxicity and has been officially recognized by the U.S. FDA (Food and Drug Administration, Silver Spring, MD, USA) as a safe (Generically Recognized as Safe (GRAS)) strain [43,44]. Furthermore, it is worth mentioning that *C. glutamicum* belongs to the group actinomycetes, which has demonstrated a close relationship to humans. This bacterium exhibits the ability to produce a diverse array of biologically active molecules and an assortment of enzymes. Notably, it surpasses bacteria and fungi in providing L-asparaginase with superior properties [45,46]. Previous research studies obtained *C. glutamicum* L-asparaginase (CgASNase) with high catalytic activity and substrate specificity which effectively inhibits the formation of acrylamide in fried potato products. Nonetheless, it suffers from poor thermal stability, which hinders its application in the food industry [40]. In this study, a combined approach of consensus design, site-point saturation mutagenesis, and combinatorial mutagenesis was employed to enhance the thermal stability of CgASNase. As a result, a double mutant with significantly improved thermal stability and no significant impact on enzyme activity was successfully generated. Meanwhile, homology modeling and a three-dimensional structure comparison analysis were utilized to investigate the molecular mechanism of the enhanced thermal stability of the mutant enzyme. Finally, the optimal treatment condition of the mutant enzyme for application in biscuits was determined using a one-way and orthogonal experimental design, aiming to achieve the maximum inhibition rate of acrylamide. These findings serve as a foundation for the industrial implementation of this enzyme.

## 2. Materials and Methods

### 2.1. Strains and Chemicals

The plasmid pET30a-*CgASNase* and the *E. coli* strain BL21 (DE3) were stored in our laboratory and used as an expression vector for the production of CgASNase and as a host for protein expression, respectively [40]. The substrate L-asparagine, the terminating agent trichloroacetic acid (TCA), and the color developer potassium mercuric iodide employed in the enzyme activity assay system were purchased from Aladdin Reagent Co., Ltd. (Shanghai, China). The plasmid extraction kits, 2×Phanta Flash Master Mix, gel purification kits, ClonExpress II One Step Cloning kits, etc., used for mutated library construction were obtained from Vazyme (Nanjing, China). All relevant primers were synthesized by GenScript (Nanjing, China). The ingredients employed in the preparation of the biscuits included gluten flour, butter, brown sugar, and salt, all of which were commercially obtainable.

### 2.2. Identification of Critical Sites for Enhancing the Thermal Stability of CgASNase

The amino acid sequence of CgASNase was submitted to the Consensus Finder online tool (http://kazlab.umn.edu, accessed on 6 December 2021) for consensus design [47]. Consensus Finder utilizes your protein sequence as a starting point, retrieves similar sequences from the NCBI database, aligns them, eliminates redundant/highly similar sequences, trims the alignments to match the size of the original query, and analyzes the consensus. The output includes a trimmed alignment, consensus sequence, frequency, and count tables for amino acids at each position, as well as a list of suggested mutations to consensus that could potentially enhance stability. To compare the amino acid sequence of CgASNase with its consensus sequence, a multiple-sequence comparison was performed using Clustal Omega online (https://www.ebi.ac.uk/Tools/msa/clustalo/, accessed on 14 September 2023). Subsequently, ESPript 3.0 (https://espript.ibcp.fr/ESPript/cgi-bin/ESPript.cgi, accessed on 14 September 2023) [48] was used for a mapping analysis aimed at identifying the key sites affecting the thermal stabilization of CgASNase.

### 2.3. Construction of Site-Point Saturation Mutations and Combinatorial Mutations

Site-point saturation mutations and combinatorial mutations were generated in accordance with the methodology described by Chi et al. [33,49]. The recombinant plasmid pET 30a-CgASNase served as the template, and mutagenesis was introduced using the whole-plasmid PCR method. All primers are shown in Appendix A.

### 2.4. High-Throughput Screening for Mutants with Enhanced Thermal Stability

The high-throughput screening method was adapted from the work of Chi et al. [33,49].

### 2.5. Expression, Purification, SDS-PAGE Analysis, and Protein Determination of L-Asparaginase

Wild-type and mutant transformants were picked out from the plates, transferred to LB liquid medium at a final concentration of 50 μg/mL of kanamycin, and incubated overnight at 37 °C, 180 rpm. Subsequently, they were transferred to fresh LB liquid medium with a final concentration of 50 μg/mL kanamycin at an inoculum of 1% and cultured at 37 °C and 180 rpm until the OD_600nm_ reached 0.6–0.8. Then, IPTG was added at a final concentration of 100 μg/mL to induce expression at 16 °C for 20 h. The cells were then harvested via centrifugation at 8000 rpm for 5 min and resuspended in Tris-HCl buffer (20 mM Tris-HCl, 300 mM NaCl, pH 8.0). The cell suspension was subsequently sonicated at 200 W for 10 min; finally, the supernatant was obtained via centrifugation at 4 °C and 10,000 rpm for 30 min. The collected supernatant was mainly purified using nickel affinity chromatography due to the presence of histidine tags in the target proteins. Heterogeneous proteins were eluted with Tris-HCl (20 mM, pH 8.0, 300 mM NaCl) containing 50 mM of imidazole, while the target proteins were eluted with Tris-HCl (20 mM, pH 8.0, 300 mM NaCl) containing 150 mM of imidazole. The molecular mass (MW), purity, and integrity of the L-asparaginase were subsequently analyzed using sodium dodecyl sulfate polyacrylamide gel electrophoresis (SDS-PAGE), and the protein concentrations of L-asparaginases were determined following the method described by Chi et al. [33,49].

### 2.6. Measurement of L-Asparaginase Activities

The activity of L-asparaginase was determined via the Nessler method, as described by Chi et al. [33,49]. The enzyme activity assay consists of a reaction system and a color development system. The former comprised 100 μL of appropriately diluted enzyme and 800 μL of PBS buffer (50 mM, pH 8.0) containing 20 mM of L-asparagine. The reaction system was incubated at 37 °C for 10 min, followed by the addition of 100 μL of 25% TCA. In the control group, the reaction was terminated by adding TCA before the start of the reaction (prior to the addition of the enzyme) while the other components remained unchanged. The latter mainly contained 40 μL of the reaction system mix and 960 μL of Nessler’s reagent dilution buffer. The absorbance was measured at OD_436nm_ after 5 min of color development. Similarly, a standard curve for enzyme activity was obtained using ammonium sulphate solution as a standard, y = 48.964x + 0.0014 (x represents the concentration of NH_4_^+^; y represents absorbance at OD_436nm_), with an R^2^ value of 0.9995. One International Unit (IU) of L-asparaginase activity was defined as the amount of enzyme required to release 1 μmol of ammonia per milliliter per minute at a pH of 8.0 and 37 °C.

### 2.7. Evaluation of Thermal Stability of CgASNase and Its Mutants

The thermal stabilities of the wild type and mutants were assessed by measuring the half-life (t_1/2_), which was determined with reference to the method of Chi et al. [33,49].

### 2.8. Enzymatic Characterization of CgASNase and Its Mutants

The enzymatic properties of an enzyme mainly include its optimum reaction temperature, optimum reaction pH, thermal stability, and kinetic parameters. Among these, the optimum reaction temperature and optimum reaction pH were determined by assessing enzyme activity at various temperatures (25–60 °C) and pH levels (pH 5.0–11.0), respectively. The highest enzyme activity was taken as 100%, and the enzyme activities at other temperatures or pHs were calculated as relative values (%). The method of thermal stability was determined as follows: first, placing a certain dilution of enzyme solution at different temperatures, extracting samples at intervals, storing them on ice, and subsequently determining the residual activity using the method of “*Measurement of L-asparaginase activities*”. The initial enzyme activity was defined as 100%, and any subsequent activity was considered residual activity (%). In addition, kinetic parameters were determined with reference to Chi et al. [33,49].

### 2.9. Molecular Modeling and Structural Analysis

To elucidate the molecular mechanisms underlying the enhanced thermal stability of the mutant enzymes, 3D structures of CgASNase and the mutant enzyme were simulated using the *Helicobacter pylori* L-asparaginase structure (PDB: 2wlt) as templates using the Swiss-Model online server. The obtained models were compared using the visualization software Pymol 2.1.0.

### 2.10. Preparation of Biscuits

The method of preparing the biscuits, with reference to Anese et al. [50,51], involved a dough recipe consisting of 100 g of flour, 40 g of butter, 35 g of brown sugar, 20 g of water, and 0.7 g of salt. The biscuits were prepared as follows: initially, the butter was liquefied through heating and then combined with the flour and brown sugar until it cooled down to room temperature. Subsequently, water with or without L-asparaginase was added to the flour. Finally, the mixture was kneaded until a smooth dough with a consistent surface formed. The dough was then placed at 4 °C for 30 min to solidify and take shape. Next, the solidified dough was pressed to a thickness of 0.3 cm and cut into round pieces using a round model. These pieces were then evenly arranged on a baking tray. Lastly, the enzyme reaction took place in a thermostat at various incubation temperatures and times. After the reaction concluded, the biscuits were baked at 230 °C for 15 min, cooled, ground, and made ready for use.

### 2.11. Application of CgASNase and Its Mutant in Biscuits

Research has indicated that enzyme dosage, enzyme reaction temperatures, and enzyme reaction times are crucial factors that influence the production of acrylamide in thermally processed foods [50,51]. Based on studies conducted by Pedreschi et al. [52] and Hendriksen et al. [53], enzyme dosages of 100, 300, 500, 700, and 900 IU/Kg, enzyme reaction times of 10, 20, 30, 40, and 50 min, and enzyme reaction temperatures of 20, 30, 40, 50, and 60 °C were selected to investigate the effect of the enzyme on the removal of acrylamide in the biscuits. In each case, a lack of enzyme addition (zero enzyme) was used as a control group. Subsequently, the aforementioned individual factors were optimized using an orthogonal experimental design (Appendix A) to determine the most effective enzyme treatment conditions. Additionally, the wild-type CgASNase and the double-mutant enzyme L42T/S213N were separately employed for dough pretreatment to assess the impact of the improved thermal stability of L-asparaginase on the effectiveness of acrylamide reduction in biscuits at the same optimal enzyme dosage, reaction time, and reaction temperature.

### 2.12. Extraction and Determination of Acrylamide

The method for extracting acrylamide from the biscuits was adapted from the work of Jiao et al. [54] with a slight modification. Firstly, 2.0 g of biscuit powder was weighed and placed into an 80 mL centrifuge tube. Then, 10 mL of ultrapure water was added, and the mixture was shaken at 200 rpm for 3 h. After centrifugation at 9000 rpm for 3 min, the supernatant was transferred to a new 80 mL centrifuge tube. This process was repeated by adding 10 mL of ultrapure water to the precipitate, and the supernatants were then collected together. Next, 10 mL of n-hexane was added to the supernatant, which was then vortexed for 10 min. After centrifuging at 9000 rpm for 3 min, the upper layer of the n-hexane phase was removed, and 10 mL of n-hexane was added again to repeat the degreasing step. The lower aqueous phase was collected as the sample for testing. Subsequently, the samples were filtered through a 0.22 µm aqueous filter membrane and stored at 4 °C. The acrylamide content in the samples to be tested was determined using high-performance liquid chromatography (HPLC) and liquid chromatography–mass spectrometry (LC-MS).

In the present study, one-way and orthogonal tests were conducted to determine the optimal treatment conditions for maximizing the removal of acrylamide from biscuits using HPLC. The HPLC conditions were as follows: the column used was an Agilent HC-C18 (4.6 × 250 mm, 5 µm); the mobile phase consisted of methanol and water in a ratio of 20:80 (*v*/*v*) with isocratic elution; the column temperature was maintained at 30 °C; the wavelength used for detection was 210 nm; the flow rate was set at 0.5 mL/min; and the injection volume was 20 μL.

Additionally, comparative experiments were performed to investigate the effect of the wild-type and mutant enzymes with improved thermal stability on acrylamide production in biscuits under optimal conditions using LC-MS. The LC-MS conditions were as follows: the chromatographic column used was a Kinetex ^®^ F5 100 A (2.1 × 100 mm, 2.6 µm, Phenomenex^TM^, Torrance, CA, USA); the samples were eluted at a 0.25 mL/min flow rate with a mixture of 90% water and 10% methanol (*v*/*v*, isometric elution) [18,55]; the electrospray ionization was performed in positive ionization mode (ESI+) and multiple-reaction monitoring (MRM) was used; the capillary voltage was set to 3.8 kV; the ion source temperature was maintained at 100 °C; the desolvent gas temperature was set as 350 °C; the desolvent gas flow rate was 10.0 L/min; and the detection ions were *m*/*z* 72.0, *m*/*z* 55.0, and *m*/*z* 44.0. The peak areas of acrylamide standard solutions at different concentrations (1, 5, 10, 50, 100, and 1000 ng/mL) were determined via LC-MS, and the standard curve was plotted with the following equation: y = 0.0085x + 0.0004 (where x represents the liquid plasma peak area and y represents the acrylamide concentration), R^2^ = 0.9987.

The acrylamide reduction refers to the ratio, expressed as a percentage, of the decrease in acrylamide content compared to the control group (zero enzyme) under different enzyme dosages, reaction temperatures, and reaction times.
$$\mathrm{Acrylamide}\;\mathrm{reduction}\left(\%\right)=\frac{C_{\mathrm{ac}}-E_{\mathrm{ac}}}{C_{\mathrm{ac}}}\times 100\%$$
where C_ac_ is the acrylamide content in the control group (zero enzyme), and E_ac_ is the acrylamide content in the experimental group.

### 2.13. Statistical Analysis

A statistical analysis was performed by repeating all experiments three times, and the results were expressed as means ± standard deviations (SDs). The significance of the data was analyzed using Duncan’s multiple tests in SPSS 13.0. *p*-values less than 0.05 were considered statistically significant and are indicated by different lowercase letters. The figures were generated using Microcalculation Origin 2021.

## 3. Results and Discussion

### 3.1. Selection of Key Sites Affecting the Thermal Stabilization of CgASNase

Protein engineering was conducted using consensus sequences as predicted mutations that align with the consensus sequence of a homologue are more likely to have neutral or positive effects. This is because natural evolution tends to eliminate amino acid residues that significantly impact protein folding and activity [56]. Often, mutations that bring a protein closer to the consensus sequence of a homologue could improve the stability of the protein and allow it to function at higher temperatures. Therefore, a strategy combining consensus design and site-point saturation mutagenesis was used to enhance the thermal stability of CgASNase. Firstly, the amino acid sequence of CgASNase was submitted to the Consensus Finder online prediction website. An analysis of the original sequence in comparison with the consensus sequence showed that a total of 136 sites were different (Figure 1). In order to increase the success rate of obtaining positive mutations, 27 mutated amino acid residues that are suspected to be stable, with a threshold value of 25% or higher (Appendix A), were selected for site-point saturation mutagenesis.

### 3.2. The Effect of Key Amino Acid Residues on the Thermal Stability of CgASNase

A total of 27 amino acid residues, namely Phe302, Thr120, Ser213, His171, Thr224, Lys294, Tyr325, Lys135, Ile67, Ile70, Phe65, Met227, His40, Ile313, Met269, Leu42, Ser251, Lys259, Glu187, Val100, Cys152, Asn71, Asn59, Asn146, Cys179, Lys208, and Gln62, were subjected to site-point saturation mutagenesis in order to obtain mutants with enhanced thermal stability (Table 1). Wild-type CgASNase and its mutants were purified via nickel column affinity chromatography and analyzed using SDS-PAGE, which revealed a single and uniformly sized band (Figure 2). Among them, the half-lives of the mutant enzymes I70L, C152I, E187N, N59S, N59Q, S213H, S213V, L42T, and H171S were found to be 1.38, 2.38, 2.60, 2.89, 3.01, 1.78, 2.02, 2.00, and 2.50 times longer, respectively, compared to the wild-type CgASNase (3.24 ± 0.23 min) at 40 °C. However, the enzyme activities of the mutant enzymes I70L, C152I, E187N, N59S, N59Q, S213H, S213V, L42T, and H171S were lower than that of CgASNase (1970.15 ± 23.54 IU/mg). Notably, the mutant enzyme L42T exhibited a 2.00-fold increase in its half-life compared to the wild type, while the enzyme activity (1922.50 ± 21.53 IU/mg) was not significantly affected (*p* > 0.05) (Table 1).

To further enhance the thermal stability of CgASNase, mutant L42T was selected as a template for site-point saturation mutagenesis at the specific sites Asn59, Ser213, Ile70, His171, Glu187, and Cys152. As a result, the double mutants L42T/S213H, L42T/S213C, L42T/N59S, and L42T/S213N with improved thermal stability were obtained. The wild-type CgASNase and its double mutants were purified via nickel affinity chromatography columns and analyzed through SDS-PAGE, which showed a single band of consistent size (Figure 3). The half-lives of the double-mutant enzymes L42T/S213H, L42T/S213N, L42T/S213C, and L42T/N59S at 40 °C were 3.72, 4.10, 2.73, and 2.68 times higher than those of the wild-type CgASNase and 1.80, 1.98, 1.32, and 1.29 times higher than the single-mutant enzyme L42T, respectively. Notably, the specific activity of the double-mutant enzyme L42T/S213N (1931.01 ± 25.84 IU/mg) did not exhibit a significant (*p* > 0.05) decrease when compared to the mutant enzyme L42T (1922.50 ± 21.53 IU/mg) or the wild type (1970.15 ± 23.54 IU/mg) (Table 2). This suggests that the double mutation further enhances the thermal stability of the enzyme without affecting its catalytic activity. The findings from these studies indicate that utilizing consensus design in combination with site-point saturation mutagenesis and combinatorial mutagenesis is an exceedingly effective strategy for designing the thermal stability of L-asparaginase.

### 3.3. Enzymatic Characteristics of the Wild-Type CgASNase and Its Mutant Enzymes

The enzymatic characteristics of both the wild type and its mutant enzymes were thoroughly examined in order to assess the potential of CgASNase and its mutant enzymes for industrial applications. This investigation took into consideration important factors such as the optimum reaction temperatures, optimum reaction pHs, and thermal stabilities of the enzyme. In comparison to the wild type, the double-mutant enzyme L42T/S213N maintained the same optimum reaction temperature of 40 °C. However, notable differences were observed: the double-mutant enzyme L42T/S213N still possessed 99% relative activity at 45 °C, and its relative activities were higher than that of the wild type across temperatures ranging from 25 to 60 °C (Figure 4A). The optimum reaction pH of the mutant enzyme L42T/S213N was identical to that of the wild type, a pH of 8.0. It is noteworthy that the relative activities of the mutant enzyme L42T/S213N were higher than those of the wild type within the pH range of 8.0 to 11.0, indicating that the mutant enzyme L42T/S213N exhibits high catalytic activity under alkaline conditions (Figure 4E). This characteristic could be applied in the production of alkaline heat-processed foods, such as soda crackers. Furthermore, the thermal stabilities of the wild-type CgASNase and the double-mutant enzyme L42T/S213N were examined at different temperatures (35, 40, and 45 °C). At 35 °C, the double-mutant enzyme L42T/S213N retained 70% residual activity after 20 min of treatment, while the wild type only retained 20% residual activity (Figure 4B). After 10 min of treatment at 40 °C, the mutant enzyme L42T/S213N still had 50% residual activity, whereas the wild type only had 10% residual activity (Figure 4C). Additionally, the mutant enzyme L42T/S213N retained 35% residual activity after 5 min of treatment at 45 °C, while the wild type completely lost its catalytic activity (Figure 4D). These results further confirm the significantly enhanced thermal stability of the double-mutant enzyme L42T/S213N. To analyze and compare the catalytic efficiency of the wild-type CgASNase and the double-mutant enzyme L42T/S213N on the substrate L-asparagine, the kinetic parameters of the two enzymes were determined (Figure 4F). The *K*_m_ and *k*_cat_ values of L42T/S213N were 4.06 ± 0.14 mM and 14,254.09 ± 23.75 min^−1^, respectively. The double-mutant enzyme L42T/S213N exhibited a higher *k*_cat_/*K*_m_ value (3125.9 min^−1^mM^−1^) compared to the wild type (2886.3 min^−1^mM^−1^), indicating that the mutation resulted in an enhancement of the enzyme’s catalytic efficiency. These findings suggest that consensus design and combinatorial mutagenesis can overcome the trade-off between thermal stability and enzyme activity, providing an effective strategy for improving the thermal stability and catalytic activity of enzymes.

### 3.4. Investigating the Molecular Mechanism of the Enhanced Thermal Stability of the Double-Mutant Enzyme L42T/S213N

In order to investigate the molecular mechanism of the enhanced thermal stability of the double-mutant enzyme L42T/S213N, intermolecular interaction forces were analyzed using three-dimensional structural simulations of the comparative analyses, which did not show any significant changes. The symmetry of the electrostatic potential on the surface of a protein is also a major factor in enhanced thermal stability, with greater symmetrical positive potentially leading to enhanced thermal stability [29,57]. Therefore, the electrostatic surfaces of the wild type and the double mutant L42T/S213N were studied. The electrostatic potential analysis of the wild type revealed that Leu42 and Ser213 were located in the neutral and negative potential regions of the CgASNase surface, respectively (Figure 5A,C). This distribution was inconsistent with the surrounding potential distribution and had a significant negative impact on overall stability. By mutating Leu to Thr at site 42, the symmetry of the surface electrostatic potential was enhanced (Figure 5B). Furthermore, the positive charge of the enzyme molecule increased after mutating Ser to Asn at site 213 (Figure 5D). Thus, the increased positive surface charge and enhanced symmetry of the surface electrostatic potential of the proteins produced by mutations are the molecular mechanisms of the enhanced thermal stability of the double-mutant enzyme L42T/S213N.

### 3.5. Application of Enzymes in Biscuits

Studies have demonstrated that enzyme dosage, enzyme reaction temperature, and enzyme reaction time are crucial factors that affect the reduction in the acrylamide content in starch-rich, thermally processed foods [50,53,58]. In order to optimize the reduction in acrylamide in biscuits using a double-mutant enzyme, ideal treatment conditions were determined through an orthogonal experimental design. The reduction in the acrylamide content served as an indicator to determine the superior level of each factor through a one-way experiment, and the results are shown in Figure 6. Firstly, the effects of different enzyme dosages on acrylamide reduction in biscuits were investigated. It was observed that when the enzyme dosage was set at 300 IU/kg flour, the acrylamide reduction in the treated biscuits was above 90% compared to those without an enzyme treatment. However, when the enzyme dosage was either below or above 300 IU/kg flour, the mitigation of acrylamide was significantly decreased (Figure 6A). The acrylamide reduction in the biscuits exhibited a tendency to increase and then decrease with an increase in the recombinant enzyme dosage, which aligns with the findings of Anese et al. [50] and Gazi et al. [58]. Subsequently, the influence of different reaction temperatures on the acrylamide content in the biscuits was explored. It was found that the highest reduction in acrylamide in the biscuits was achieved when the dough was maintained at 40 °C (Figure 6B), which is consistent with the optimum reaction temperature for the enzyme based on its enzymatic properties. Furthermore, when the temperature was low, the exchange rate of water molecules in the dough was reduced, which can lead to a decrease in the reaction rate of the enzyme. Conversely, when the temperature was excessively high, the evaporation of water in the dough resulted in an uneven distribution of water within the dough, impeding the enzyme from fully hydrolyzing L-asparagine in the dough and consequently reducing the decrease in acrylamide [58]. Lastly, the impact of enzymatic reaction times on the acrylamide content in the biscuits was examined. It was observed that when the reaction time was less than 30 min, the increase in acrylamide reduction became more pronounced with longer reaction times (Figure 6C). Gazi et al. [58] discovered that different enzymatic reaction times led to significant changes in the acrylamide content in biscuits, even if the difference in L-asparagine content between two sets of doughs was minimal. Therefore, it is imperative to ensure an adequate enzymatic reaction time. When the reaction time exceeds 40 min, each additional 10 min increment only enhances the removal of acrylamide by approximately 1%. Consequently, 40 min represents the optimal reaction time.

To exclude possible interactions between factors, a three-factor, three-level L9 (3^4^) orthogonal experimental design was used (Appendix A). The magnitude of the extreme difference (R) reflects the level of significance of different factors. As can be seen from Appendix A, the three treatment factors affected the acrylamide content reduction in the order of precedence A—enzyme dosage > B—enzyme reaction times > C—enzyme reaction temperatures, with enzyme dosage exerting the most substantial impact on the reduction in acrylamide. An analysis of variance (ANOVA) on the results of the orthogonal experiments also confirmed the significant (*p* < 0.05) influence of enzyme dosage on the acrylamide reduction by L-asparaginase (Appendix A), which is similar to the findings of Anese et al. [50].

The theoretical optimal treatment condition A2B3C2 and the actual better treatment condition A2B1C2 were obtained via the orthogonal experiment. The results of the orthogonal experiment were further validated. The ANOVA analysis of the orthogonal experiments revealed that the enzymatic reaction times had no significant impact on reducing the acrylamide content. To further investigate the effect of enzymatic reaction times on acrylamide reduction, the combination A2B2C2 was included in the verification process. The results indicate that the enzymatic reaction times had no significant effect on the acrylamide content in the biscuits (*p* > 0.05) (Appendix A), which aligned with the findings of the one-factor experiment. Consequently, the optimal treatment condition was determined to be A2B1C2, encompassing an enzyme dosage of 300 IU/kg, an enzyme reaction time of 30 min, and an enzyme reaction temperature of 40 °C.

To assess the impact of the enhanced thermal stability of L-asparaginase on the reduction in acrylamide in biscuits, the wild-type CgASNase and the double-mutant enzyme L42T/S213N were used to pre-treat the dough under the same and optimal treatment conditions. After baking, acrylamide was extracted, and its content in the biscuits was determined using LC-MS. The results, as shown in Table 3, indicated that the content of acrylamide in the control group (biscuits without enzyme treatment) was 464.74 ± 6.68 μg/kg. However, when treated with the wild-type CgASNase, the content of acrylamide decreased to 145.11 ± 4.19 μg/kg, resulting in a reduction of 68.78%. On the other hand, when treated with the double-mutant enzyme L42T/S213N, the content of acrylamide further decreased to 68.30 ± 1.67 μg/kg, leading to an impressive reduction of 85.31% (Table 3). The results of this study show that the double-mutant enzyme had similar enzymatic properties to the wild type. Therefore, these findings clearly demonstrate that the mutant enzyme with enhanced thermal stability has the ability to significantly decrease the acrylamide content in biscuits. Gazi et al. [58] investigated the effect of the enzyme dosage, dough resting times and temperatures, mixing speeds and times, and recipe components on acrylamide reduction in thin, crisp biscuits and determined the optimal treatment condition under which the acrylamide reduction reached 80%, with a lower inhibition than that of the mutant enzyme L42T/S213N. These results suggest that the double-mutant enzyme L42T/S213N has great potential for application in the food industry.

## 4. Conclusions

Compared to the methods of directed evolution and rational design, semi-rational design is a simple, effective, and highly successful approach. Of these, the consensus design strategy proves to be effective in enhancing the thermal stability of proteins. In this study, the consensus design strategy was employed to address the issue of poor thermal stability in CgASNase. The key sites influencing the thermal stability of CgASNase were identified, and site-point saturation mutation and combinatorial mutation techniques were used to screen the double-point-mutant enzyme L42T/S213N, which exhibited a significant improvement in thermal stability without compromising enzyme activity, as its half-life at 40 °C increased by 3.1-fold compared to the wild type. Three-dimensional structural simulations and structure comparison analyses revealed that the enhanced thermal stability of the double-mutant enzyme was attributed to an increase in positive surface charge and a more symmetrical positive potential on the protein surface. Furthermore, to investigate the acrylamide-removal capability of the mutant enzyme L42T/S213N in thermally processed foods, an optimal treatment condition was determined through one-way and orthogonal experimental designs. The optimal condition included an enzyme dosage of 300 IU/kg, an enzyme reaction temperature of 40 °C, and an enzyme reaction time of 30 min. Under these conditions, compared to wild-type CgASNase, the double-mutant enzyme L42T/S213N showed a significant increase in acrylamide reduction, achieving a reduction rate of up to 85.31%. Therefore, the double-mutant enzyme L42T/S213N holds great promise as a candidate for reducing the acrylamide content in starch-rich, thermally processed foods.

Although this study obtained a significantly improved CgASNase with respect to thermostability which can also reduce the acrylamide content in baked foods, the food industry requires a more thermally stable L-asparaginase. Therefore, future research can build upon this foundation by employing artificial intelligence or machine learning methods to further design CgASNase and obtain mutated enzymes with superior performance in terms of thermal stability, catalytic efficiency, and substrate specificity. Additionally, considering the cost of industrial applications, it is possible to enhance the enzyme’s reusability, catalytic efficiency, and thermal stability by covalently immobilizing it onto the surface of magnetic nanoparticles. Furthermore, studying the enzyme in a bioreactor can increase its production yield, making it more readily applicable to various industrial processes.

## Figures and Tables

**Figure 1 foods-12-04364-f001:**
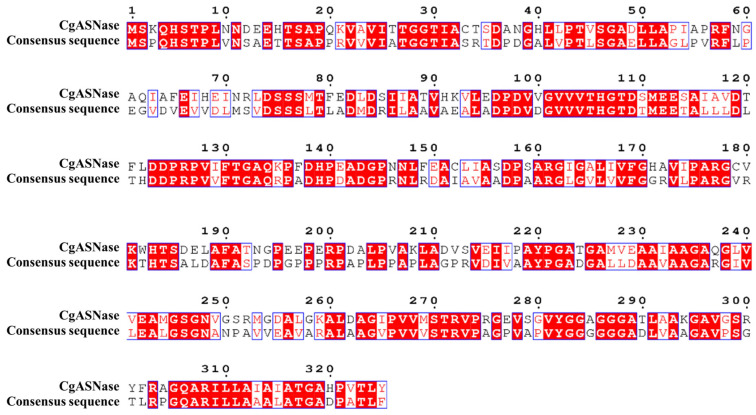
Multiple sequence alignments of CgASNase with its consequence sequence. Conserved residues are designated with red squares and white lettering, and similar residues are designated with red lettering.

**Figure 2 foods-12-04364-f002:**
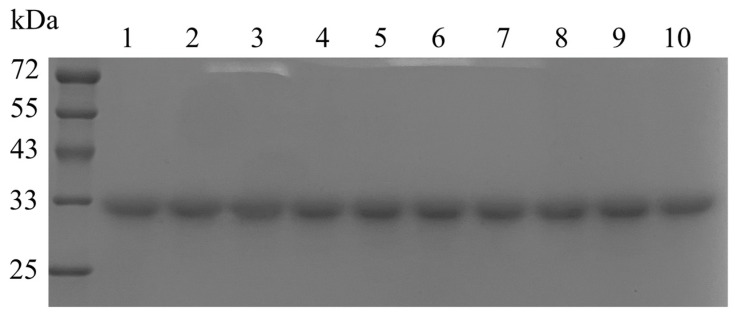
SDS-PAGE analysis of wild-type CgASNase and its mutants. M: protein marker; lane 1: wild-type CgASNase; lane 2: I70L; lane 3: C152I; lane 4: E187N; lane 5: N59S; lane 6: N59Q; lane 7: S213H; lane 8: S213V; lane 9: L42T; lane 10: H171S.

**Figure 3 foods-12-04364-f003:**
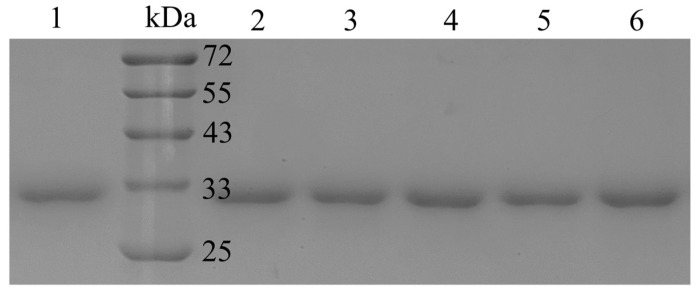
SDS-PAGE analysis of wild-type CgASNase and its mutants. M: Protein marker; lane 1: wild-type CgASNase; lane 2: L42T; lane 3: L42T/S213H; lane 4: L42T/S213N; lane 5: L42T/S213C; lane 6: L42T/N59S.

**Figure 4 foods-12-04364-f004:**
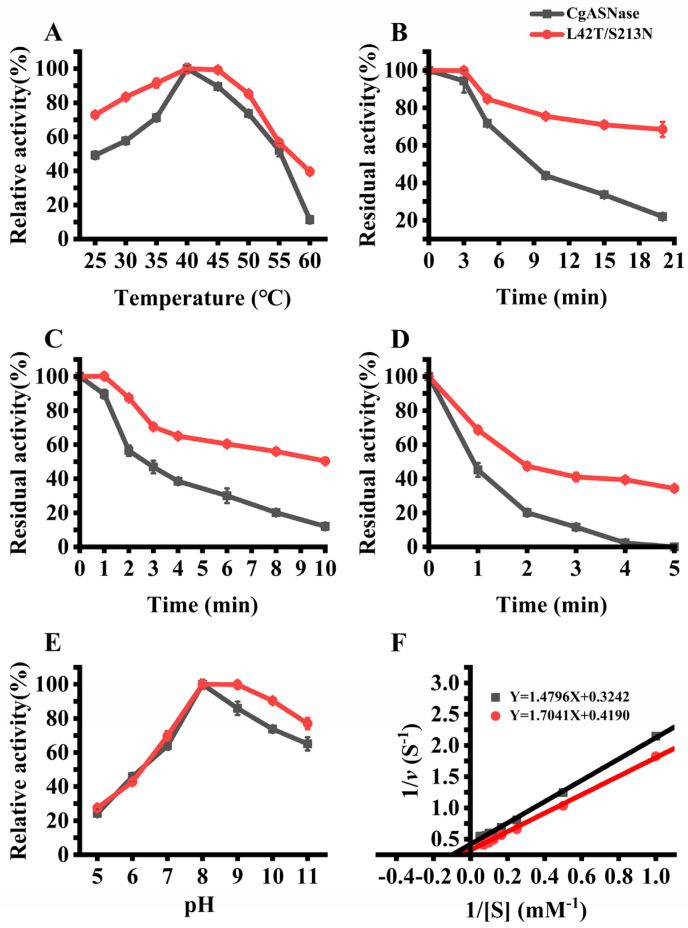
Enzymatic properties of wild-type CgASNase and its double-mutant enzyme L42T/S213N. (**A**) Optimum reaction temperatures of wild-type CgASNase and its double-mutant enzyme L42T/S213N; (**B**) thermal stability of wild-type CgASNase and its double-mutant enzyme L42T/S213N at 35 °C; (**C**) thermal stability of wild-type CgASNase and its double-mutant enzyme L42T/S213N at 40 °C; (**D**) thermal stability of wild-type CgASNase and its double-mutant enzyme L42T/S213N at 45 °C; (**E**) optimum reaction pH of wild-type CgASNase and its double-mutant enzyme L42T/S213N; (**F**) kinetic parameters of wild-type CgASNase and its double-mutant enzyme L42T/S213N.

**Figure 5 foods-12-04364-f005:**
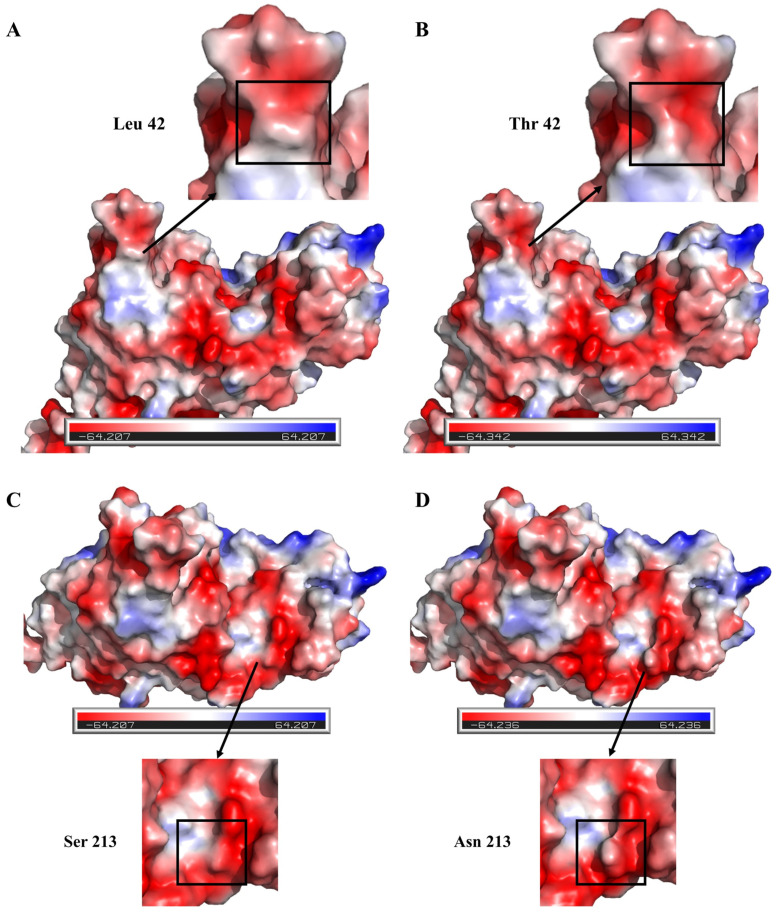
Surface electrostatic potential maps of wild-type CgASNase and double-mutant enzyme L42T/S213N. (**A**) Surface electrostatic potential plot of wild-type CgASNase at site 42; (**B**) surface electrostatic potential plot of mutant enzyme L42T at site 42; (**C**) surface electrostatic potential plot of wild-type CgASNase at site 213; (**D**) surface electrostatic potential plot of mutant enzyme S213N at site 213. Positive, negative, and neutral values of the electrostatic potential are indicated in blue, red, and white, respectively.

**Figure 6 foods-12-04364-f006:**
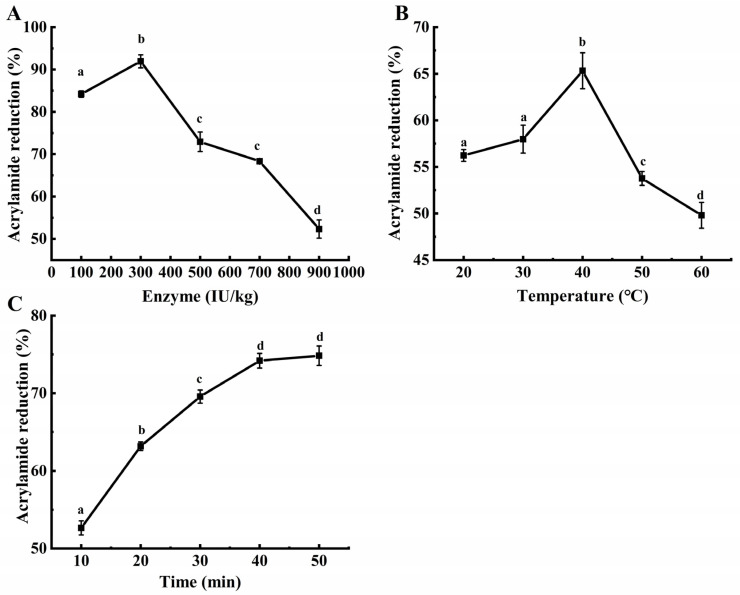
Effects of different enzyme dosage (**A**), enzyme reaction temperature (**B**), and enzyme reaction time (**C**) on reducing acrylamide in biscuits.

**Table 1 foods-12-04364-t001:** Specific activities and half-lives of wild-type CgASNase and its single mutants.

Enzymes	Specific Activities (IU/mg)	Half-Lives (40 °C, min)
Wild type	1970.15 ± 23.54 ^a^	3.24 ± 0.23 ^f^
I70L	1661.67 ± 25.31 ^c^	4.47 ± 0.42 ^e^
C152I	1354.95 ± 13.59 ^d^	7.71 ± 0.41 ^b^
E187N	1211.36 ± 12.75 ^e^	8.42 ± 0.19 ^b^
N59S	1206.90 ± 21.65 ^e^	9.36 ± 0.53 ^a^
N59Q	1124.77 ± 32.76 ^f^	9.75 ± 0.38 ^a^
S213H	966.94 ± 13.92 ^g^	5.77 ± 0.45 ^c^
S213V	1740.39 ± 23.82 ^b^	6.54 ± 0.25 ^d^
L42T	1922.50 ± 21.53 ^a^	6.71 ± 0.61 ^d^
H171S	1613.30 ± 31.06 ^c^	8.10 ± 0.54 ^b^

**Table 2 foods-12-04364-t002:** Specific activities and half-lives of wild-type CgASNase, single mutant L42T, and its double mutants.

Enzymes	Specific Activities (IU/mg)	Half-Lives (40 °C, min)
Wild type	1970.15 ± 23.54 ^a^	3.24 ± 0.23 ^d^
L42T	1922.50 ± 21.53 ^a^	6.71 ± 0.61 ^c^
L42T/S213H	1627.86 ± 32.67 ^c^	12.05 ± 0.56 ^a^
L42T/S213N	1931.01 ± 25.84 ^a^	13.29 ± 0.91 ^a^
L42T/S213C	1763.49 ± 64.24 ^b^	8.84 ± 0.73 ^b^
L42T/N59S	1491.18 ± 36.57 ^d^	8.70 ± 0.82 ^b^

**Table 3 foods-12-04364-t003:** Effects of wild-type CgASNase and its double mutant L42T/S213N on the inhibition of acrylamide in biscuits.

Groups	Acrylamide Content (μg/kg)	Acrylamide Reduction (%)
Control group	464.74 ± 6.68 ^a^	-
CgASNase	145.11 ± 4.19 ^b^	68.78 ± 2.73 ^a^
L42T/S213N	68.30 ± 1.67 ^c^	85.31 ± 1.01 ^b^

## Data Availability

The data used to support the findings of this study can be made available by the corresponding author upon request.

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
