# Peer review of "Thermal Stability Enhancement of L-Asparaginase from *Corynebacterium glutamicum* Based on a Semi-Rational Design and Its Effect on Acrylamide Mitigation Capacity in Biscuits"

_foods, 2023, doi:10.3390/foods12234364_

Round 1

Reviewer 1 Report

Comments and Suggestions for Authors

The research findings have a significant contribution into food industrial application to the increasing potential of enzymatic elimination of acrylamide in baked foods.

In Materials and Methods (Chapter 2.12) there are some discrepancies for acrylamide analysis regarding detection technique (MS) and reported detector parameter (wavelength of 210 nm). There is a need to correct it. 

Author Response

Comments:

The research findings have a significant contribution into food industrial application to the increasing potential of enzymatic elimination of acrylamide in baked foods.

Response: We appreciate the referee's support and encouragement for our work. Thanks for these great suggestions. Below are our detailed point-to-point responses.

Point 1: In Materials and Methods (Chapter 2.12) there are some discrepancies for acrylamide analysis regarding detection technique (MS) and reported detector parameter (wavelength of 210 nm). There is a need to correct it.

Response 1: We appreciate the referee's valuable suggestion and sincerely apologize for any confusion arising from our description. In order to enhance clarity and comprehension, we have made revisions to the manuscript for Chapter 2.12 (p. 6, line 267-276). Additionally, we have included supplementary information pertaining to LC/MS on page 6, lines 277-279, and have provided the corresponding references in the revised version.

Line 267-276: In the present study, one-way and orthogonal tests were conducted to determine the optimal treatment conditions for maximizing the removal of acrylamide from biscuits using HPLC. The HPLC conditions were as follows: the column used was Agilent HC-C18 (4.6×250 mm, 5 µm); the mobile phase consisted of methanol and water in a ratio of 20:80 (V/V) with isocratic elution; the column temperature was maintained 30 ℃; the wavelength used for detection was 210 nm; the flow rate was set at 0.5 mL/min; and the injection volume was 20 μL.

Additionally, comparative experiments were performed to investigate the effect of wild type and mutant enzymes with improved thermal stability on acrylamide production in biscuits under the optimal conditions using LC-MS. The LC-MS conditions were as follows:

Line 277-279: the chromatographic column used was Kinetex ® F5 100 A (2.1 × 100 mm, 2.6 µm); the samples were eluted at a 0.25 mL/min flow rate with a mixture of 90% water and 10% methanol (v/v, isometric elution) [18, 55]

Reference:

  1. Chi, H.; Chen, M.; Jiao, L.; Lu, Z.; Bie, X.; Zhao, H.; Lu, F., Characterization of a novel L-asparaginase from Mycobacterium gordonae with acrylamide mitigation potential. Foods 2021, 10, (11), 2819.
  2. Niu, J.; Yan, R.; Shen, J.; Zhu, X.; Meng, F.; Lu, Z.; Lu, F., Cis-element engineering promotes the expression of Bacillus subtilis type I L-asparaginase and its application in food. International Journal of Molecular Sciences 2022, 23, (12), 6588.

Reviewer 2 Report

Comments and Suggestions for Authors

Reviewer report for paper entitled "Thermal stability enhancement of L-asparaginase from Corynebacterium glutamicum based on a semi-rational design and its effect on acrylamide mitigation capacity in biscuits". This research is high interest for both scientific society with high interest for food industries. However, some points need to be addressed before the publication of this work as follows:

- In introduction part, need to highlight in more depth the underlying mechanism of L-asparaginase for detoxification of acrylamide. and also why L-asparaginase from C. glutamicum is more suitable for this process. 

- In materials and methods part, authors report on using recombinant E. coli BL21 with L-asparaginase encoded plasmid, however, only purification method was reported. Authors, need to add the cultivation condition for L-asparaginase production in this recombinant strain and induction process during cultivation. 

- In Figure 6A, authors need to provide the point of (zero enzyme) in the curve as negative control for this experiment. 

- Conclusion part is well written, however, I recommend to add another part related to future prospectives, and future work suggested for full industrialization. Limitation of this work need to be addressed such as cost and other factors. 

In general, this work is of high quality and in the journal scope.

Comments on the Quality of English Language

acceptable 

Author Response

Comments:

Reviewer report for paper entitled "Thermal stability enhancement of L-asparaginase from Corynebacterium glutamicum based on a semi-rational design and its effect on acrylamide mitigation capacity in biscuits". This research is high interest for both scientific society with high interest for food industries. In general, this work is of high quality and in the journal scope. However, some points need to be addressed before the publication of this work as follows:

Response: We appreciate the referee’s enthusiasm for our work. Thanks for these great suggestions. Below are our detailed point-to-point responses.

Point 1: In introduction part, need to highlight in more depth the underlying mechanism of L-asparaginase for detoxification of acrylamide. and also why L-asparaginase from C. glutamicum is more suitable for this process.

Response 1: Thanks for your suggestion. We apologize for the lack of a comprehensive explanation regarding the process of acrylamide elimination by L-asparaginase. We have included this information in the Introduction section (p. 2, line 50-57). Furthermore, for L-asparaginase to be applicable in the food industry, it must exhibit high catalytic activity, thermal stability, substrate specificity, and originate from a safe (GRAS) source. In contrast, Corynebacterium glutamicum is a secure strain for industrial production. It does not produce endotoxicity and is generally recognized as safe (GRAS) by the US FDA (Sains Malaysiana 2023, 52, (2), 431-439; Biotechnology and Bioprocess Engineering 2023, 28, (3), 419-427). Consequently, this strain is extensively employed in the food industry. Additionally, C. glutamicum belongs to the actinomycetes group, which has been demonstrated its close relation to humans. This group is capable of generating a diverse range of biologically active molecules and enzymes. As a result, it can provide L-asparaginase with superior attributes compared to bacteria and fungi (European Journal of Pharmacology 2016, 771, 199-210; Gene 2016, 590, (2), 220-226). Previous research has indicated that C. glutamicum L-asparaginase exhibits high catalytic activity and exclusively acts upon the substrate L-asparagine. It also possesses the ability to inhibit the formation of acrylamide in fried crisps. However, its thermal stability is inadequate. To address this limitation, the enzyme was subjected to molecular modification to enhance its thermal stability. The objective was to obtain a mutant enzyme of C. glutamicum L-asparaginase with improved thermal stability, rendering it more suitable for industrial production. We have also added relevant studies in the Introduction section (p. 3, line 115-123).

line 50-57: L-asparaginase is a hydrolytic enzyme that facilitates the hydrolysis of L-asparagine, resulting in the formation of L-aspartic acid and ammonia [8]. The occurrence of acrylamide in food products is primarily attributed to the Maillard reaction between L-asparagine and reducing sugars present in raw materials when subjected to temperatures exceeding 120 °C. However, it is worth noting that the product L-aspartic acid does not participate in the Maillard reaction [9, 10]. Consequently, L-asparaginase is predominantly employed to control the formation of acrylamide in food at the source by diminishing the content of L-asparagine, which is a prerequisite substance for acrylamide formation [11].

line 115-123: Corynebacterium glutamicum is extensively used in the food industry as an industrially produced strain with a remarkable level of safety. It is noteworthy that this strain does not generate endotoxicity and has been officially recognized by the US FDA (Food and Drug Administration) as a safe (Generically Recognized as Safe, GRAS) strain [43, 44]. Furthermore, it is worth mentioning that C. glutamicum belongs to the group of actinomycetes, which has demonstrated a close relationship to humans. This bacterium exhibits the ability to produce a diverse array of biologically active molecules and an assortment of enzymes. Notably, it surpasses bacteria and fungi in providing L-asparaginase with superior properties [45, 46].

Reference:

  1. Xu, F.; Oruna-Concha, M.-J.; Elmore, J. S., The use of asparaginase to reduce acrylamide levels in cooked food. Food Chemistry 2016, 210, 163-171.
  2. Liu, Y.; Wang, P.; Chen, F.; Yuan, Y.; Zhu, Y.; Yan, H.; Hu, X., Role of plant polyphenols in acrylamide formation and elimination. Food Chemistry 2015, 186, 46-53.
  3. Jia, R.; Wan, X.; Geng, X.; Xue, D.; Xie, Z.; Chen, C., Microbial L-asparaginase for application in acrylamide mitigation from food: Current research status and future perspectives. Microorganisms 2021, 9, (8), 1659.
  4. Shahana Kabeer, S.; Francis, B.; Vishnupriya, S.; Kattatheyil, H.; Joseph, K. J.; Krishnan, K. P.; Mohamed Hatha, A. A., Characterization of L-asparaginase from Streptomyces koyangensis SK4 with acrylamide-minimizing potential in potato chips. Brazilian journal of microbiology 2023, 54, (3), 1645-1654.
  5. Ruslan, U. S.; Ahmad Raston, N. H.; Mohd Sharif, N. A.; Neoh, H. M.; Nathan, S.; Ramzi, A. B., Development of Corynebacterium glutamicum as staphylococcal-targeting chassis via the construction of autoinducing peptide (aip)-responsive expression system. Sains Malaysiana 2023, 52, (2), 431-439.
  6. Jeon, E. J.; Lee, Y.-M.; Choi, E. J.; Kim, S.-B.; Jeong, K. J., Production of tagatose by whole-cell bioconversion from fructose using Corynebacterium glutamicum. Biotechnology and Bioprocess Engineering 2023, 28, (3), 419-427.
  7. Ali, U.; Naveed, M.; Ullah, A.; Ali, K.; Shah, S. A.; Fahad, S.; Mumtaz, A. S., L-asparaginase as a critical component to combat Acute Lymphoblastic Leukaemia (ALL): A novel approach to target ALL. European Journal of Pharmacology 2016, 771, 199-210.
  8. Meena, B.; Anburajan, L.; Vinithkumar, N. V.; Shridhar, D.; Raghavan, R. V.; Dharani, G., Molecular expression of L-asparaginase gene from Nocardiopsis alba NIOT-VKMA08 in Escherichia coli: A prospective recombinant enzyme for leukaemia chemotherapy. Gene 2016, 590, (2), 220-226.

Point 2: In materials and methods part, authors report on using recombinant E. coli BL21 with L-asparaginase encoded plasmid, however, only purification method was reported. Authors, need to add the cultivation condition for L-asparaginase production in this recombinant strain and induction process during cultivation.

Response 2: Thanks for the referee’s suggestion and sorry for our mistake, the cultivation condition for L-asparaginase production has been added in the revised manuscript (p. 4, line 171-181).

Line 171-181: Wild-type and mutant transformants were picked out from the plates, transferred to LB liquid medium at a final concentration of 50 μg/mL kanamycin, and incubated overnight at 37°C, 180 rpm. Subsequently, they were transferred to fresh LB liquid medium with a final concentration of 50 μg/mL kanamycin at an inoculum of 1% and cultured at 37°C, 180 rpm until the OD600 nm reached to 0.6-0.8. Then, IPTG was added at a final concentration of 100 μg/mL to induce expression at 16°C for 20 h. The cells were then harvested by centrifugation at 8,000 rpm for 5 min and resuspended in Tris-HCl buffer (20 mM Tris-HCl, 300 mM NaCl, pH 8.0). The cell suspension was subsequently sonicated at 200W for 10 min, and finally, the supernatant was obtained by centrifugation at 4°C and 10,000 rpm for 30 min.

Point 3: In Figure 6A, authors need to provide the point of (zero enzyme) in the curve as negative control for this experiment.

Response 3: Thanks for the referee’s suggestion. We apologize for not clearly defining the concept of acrylamide reduction in the Materials and Methods section. In this study, the control group was set without the addition of enzymes (zero enzyme). The acrylamide reduction refers to the ratio, expressed as a percentage, of the decrease in acrylamide content compared to the control group (zero enzyme) under different enzyme addition amounts, reaction temperatures, and reaction times. Therefore, all results in Figure 6 are calculated and analyzed on the basis of the control group (zero enzyme). The relevant definitions and calculation formulas have been added to the Materials and Methods section (p. 5-6, line 246-247; p. 6, line 288-293).

Line 246-247: In all of them, no enzyme addition (zero enzyme) was used as the control group.

Line 288-293: The acrylamide reduction refers to the ratio, expressed as a percentage, of the decrease in acrylamide content compared to the control group (zero enzyme) under different dosage, reaction temperatures, and reaction times.

where Cac is acrylamide content in the control group (zero enzyme), Eac is acrylamide content in the experimental group.

Point 4: Conclusion part is well written, however, I recommend to add another part related to future prospectives, and future work suggested for full industrialization.

Response 4: Thanks for the referee’s advice. We have added future prospectives in the Conclusion section (p. 14-15, line 530-540).

Line 530-540: Although this study has already obtained a significant thermostability improved CgASNase, which can also reduce the acrylamide content in baked foods, the food industry requires a more thermally stable L-asparaginase. Therefore, future research can build upon this foundation by employing artificial intelligence or machine learning methods to further design CgASNase and obtain mutated enzymes with superior performance in terms of thermal stability, catalytic efficiency, and substrate specificity. Additionally, considering the cost of industrial applications, it is possible to enhance the enzyme's reusability, catalytic efficiency, and thermal stability by covalently immobilizing it onto the surface of magnetic nanoparticles. Furthermore, studying the enzyme in a bioreactor can increase its production yield, making it more readily applicable to various industrial processes.

Point 5: Limitation of this work need to be addressed such as cost and other factors.

Response 5: Thanks for the referee’s suggestion. We have discussed this information in the Conclusion section (p. 14, line 530-540).

Reviewer 3 Report

Comments and Suggestions for Authors

The manuscript reports the rational mutagenesis of microbial asparaginase to be employed in the bakery industry to address the problem of acrylamide production during food baking.

In particular, efforts were made to improve the enzyme's thermal stability and catalytic activity.

Given the specific application of such an enzyme and the food manufacturing process reported, the authors should explain and discuss

1) the advantages of engineering an enzyme rather than mining thermostable asparaginases, for example from thermophilic bacteria

2) the reason why a thermostable enzyme is better than a cold-active one, given that in the specific case, the manufacturing process includes a refrigeration step, during which the reaction may occur.

Moreover, it is not clear if the observed improvement in asparagine reduction is due to the extended enzyme's half-life or to the enhanced enzymatic activity (as a principle, these two concepts might not be related to each other; however it must be clarified, given that the manuscript is focused on the improvement of thermal stability).

Comments on the Quality of English Language

Overall, the English language is fine, but some points to fix are present throughout the text (e.g. line 56).

Author Response

Comments:

The manuscript reports the rational mutagenesis of microbial asparaginase to be employed in the bakery industry to address the problem of acrylamide production during food baking. In particular, efforts were made to improve the enzyme's thermal stability and catalytic activity. Given the specific application of such an enzyme and the food manufacturing process reported, the authors should explain and discuss.

Response: We appreciate the referee’s great comments to improve the quality and impact of our work. We have performed suggested revisions. Below are our detailed point-to-point responses.

Point 1: the advantages of engineering an enzyme rather than mining thermostable asparaginases, for example from thermophilic bacteria.

Response 1: Thanks for referee’s suggestion. We have added the explanation and discussion of the advantages of engineering an enzyme rather than mining thermostable asparaginases to the Introduction section (p. 2, line 84-93).

Line 84-93: Currently, there are two main approaches to obtaining L-asparaginase with high thermal stability. The first is to screen for new L-asparaginase genes from extreme environments, while the second involves protein engineering to reconstruct mesophilic L-asparaginase [28, 29]. Although several thermophilic L-asparaginases have been extracted from hot springs and hydrothermal vents (such as Thermococcus and Pyrocococcus sp.), their enzymatic activity and substrate specificity do not meet the requirements for industrial applications [30, 31]. Moreover, the screening process for these enzymes is time-consuming and costly [32]. Conversely, these limitations can be overcome through protein engineering (irrational design (directed evolution), semi-rational design, and rational design).

Reference:

  1. Zhang, X.; Wang, Z.; Wang, Y.; Li, X.; Zhu, M.; Zhang, H.; Xu, M.; Yang, T.; Rao, Z., Heterologous expression and rational design of L-asparaginase from Rhizomucor miehei to improve thermostability. Biology (Basel) 2021, 10, (12), 1346.
  2. Long, S.; Zhang, X.; Rao, Z.; Chen, K.; Xu, M.; Yang, T.; Yang, S., Amino acid residues adjacent to the catalytic cavity of tetramer L-asparaginase II contribute significantly to its catalytic efficiency and thermostability. Enzyme and Microbial Technology 2016, 82, 15-22.
  3. Dumina, M.; Zhgun, A.; Pokrovskaya, M.; Aleksandrova, S.; Zhdanov, D.; Sokolov, N.; El'darov, M., A novel L-asparaginase from hyperthermophilic archaeon Thermococcus sibiricus: Heterologous expression and characterization for biotechnology application. International Journal of Molecular Sciences 2021, 22, (18), 9894.
  4. Li, X.; Zhang, X.; Xu, S.; Xu, M.; Yang, T.; Wang, L.; Zhang, H.; Fang, H.; Osire, T.; Rao, Z., Insight into the thermostability of thermophilic L-asparaginase and non-thermophilic L-asparaginase II through bioinformatics and structural analysis. Applied Microbiology and Biotechnology 2019, 103, (17), 7055-7070.
  5. Kamble, A.; Srinivasan, S.; Singh, H., In-silico bioprospecting: Finding better enzymes. Molecular Biotechnology 2019, 61, (1), 53-59.

Point 2: the reason why a thermostable enzyme is better than a cold-active one, given that in the specific case, the manufacturing process includes a refrigeration step, during which the reaction may occur.

Response 2: Thanks for referee’s suggestion. The catalytic activity of cold-active enzymes under low-temperature conditions is achieved through the flexibility of their structure. Cold-active enzymes exhibit a higher proportion of α-helices and a lower proportion of β-folded secondary structures. Since these β-folds tend to form more rigid structures, the higher content of α-helices in cold-active enzymes allows for greater flexibility at low temperatures. Increased flexibility is the main characteristic contributing to the low-temperature catalytic activity of these enzymes. However, this increased flexibility also leads to decreased stability, making cold-active enzymes sensitive to heat as temperatures rise, even at room temperature. While lower temperatures can reduce contamination from mesophilic bacteria during food processing, the unique nature of cold-active enzymes results in poor storage stability in industrial applications (Frontiers in Microbiology 2023, 14, 1152847; Bioprocess and Biosystems Engineering 2023, 46, (10), 1399-1410; marine drugs 2010, 21, (332)). Considering the issue of storage period in the industrial application of enzyme preparations, the development of highly efficient and stable artificial cold-active enzymes holds great promise.

However, thermophilic enzymes exhibit remarkable stability under high temperature conditions and possess strong resistance to organic solvents and denaturants. Thermophilic enzymes hold significant and potential application value in various fields such as the food industry, environmental protection, energy, and petroleum extraction. Utilizing thermophilic enzymes as biocatalysts offers the following advantages: (1) Low preparation cost of enzyme preparations due to the high stability of thermophilic enzymes, allowing for purification and packaging at room temperature while maintaining long-lasting activity. (2) Acceleration of kinetic reactions as reaction temperature increases, resulting in enhanced enzymatic catalytic ability due to increased molecular movement speed. (3) Reduced requirements for reactor cooling systems, leading to decreased energy consumption. The heat-resistant nature of thermophilic enzymes eliminates the need for complex cooling devices during production (Journal of Genetic Engineering and Biotechnology (2023) 21:37; Journal of Basic Microbiology 2021;1-14; Bioresource Technology 89 (2003) 17-34). This not only saves expenses but also reduces the environmental pollution caused by the cooling process.

Therefore, a thermostable enzyme holds significant and valuable potential for application in the food industry. We have added relevant discussion to the Introduction section (p. 2, line 75-84).

Line 75-84: Enzymes that possess the desired thermal stability exhibit notable advantages in terms of accelerating reactions, increasing evolutionary potential, reducing microbial contamination, and lowering production costs [21-24]. The catalytic activity of cold-active enzymes at low temperatures is achieved through the flexibility of their structure. However, the increased flexibility also leads to a decrease in the stability of these enzymes. As the temperature rises, even reaching room temperature, cold-active enzymes become sensitive to heat, resulting in poor storage stability and making them difficult to meet industrial applications [25-27]. Therefore, a thermostable enzyme holds significant and valuable potential for application in the food industry.

References:

  1. Bloom, J. D.; Labthavikul, S. T.; Otey, C. R.; Arnold, F. H., Protein stability promotes evolvability. Proceedings of the national academy of sciences of the united states of america 2006, 103, (15), 5869-5874.
  2. Che Hussian, C. H. A.; Leong, W. Y., Thermostable enzyme research advances: a bibliometric analysis. Journal of Genetic Engineering and Biotechnology 2023, 21, (1), 37.
  3. Arbab, S.; Ullah, H.; Khan, M. I. U.; Khattak, M. N. K.; Zhang, J.; Li, K.; Hassan, I. U., Diversity and distribution of thermophilic microorganisms and their applications in biotechnology. Journal of Basic Microbiology 2022, 62, (2), 95-108.
  4. Haki, G. D.; Rakshit, S. K., Developments in industrially important thermostable enzymes: a review. Bioresource Technology 2003, 89, (1), 17-34.
  5. Liu, Y.; Zhang, N.; Ma, J.; Zhou, Y.; Wei, Q.; Tian, C.; Fang, Y.; Zhong, R.; Chen, G.; Zhang, S., Advances in cold-adapted enzymes derived from microorganisms. Frontiers in Microbiology 2023, 14, 1152847.
  6. Liu, Y.; Jia, K.; Chen, H.; Wang, Z.; Zhao, W.; Zhu, L., Cold-adapted enzymes: mechanisms, engineering and biotechnological application. Bioprocess and Biosystems Engineering 2023, 46, (10), 1399-1410.
  7. Wang, J.; Zhu, M.; Wang, P.; Chen, W., Biochemical properties of a cold-active chitinase from marine Trichoderma gamsii R1 and its application to preparation of chitin oligosaccharides. marine drugs 2010, 21, (332).

Point 3: Moreover, it is not clear if the observed improvement in asparagine reduction is due to the extended enzyme's half-life or to the enhanced enzymatic activity (as a principle, these two concepts might not be related to each other; however it must be clarified, given that the manuscript is focused on the improvement of thermal stability.

Response 3: Thanks for referee’s advice. For our mistake, the conditions for the pretreatment of flour with wild type and double mutant enzyme were not clearly stated in the Materials and Methods. It is essential to have the same enzyme dosage, reaction time, and reaction temperature in order to evaluate the enhanced thermal stability leading to the higher acrylamide reduction in biscuits. Additionally, the enzymatic properties indicate that wild type and double mutant L42T/S213N have similar characteristics, thereby ruling out other factors contributing to the acrylamide reduction. In conclusion, the significant improvement in thermal stability of the double mutant enzyme is the reason for the enhanced acrylamide reduction. The corresponding explanations have also been added to the Materials and Methods (p. 6, line 249-252), as well as the Results and Discussion sections (p. 13, line 488-490; p. 14, line 497-500).

Line 249-252: Additionally, the wild type CgASNase and double mutant enzyme L42T/S213N were separately employed for dough pretreatment to assess the impact of improved thermal stability of L-asparaginase on the effectiveness of acrylamide reduction in biscuits at the same optimal enzyme dosage, reaction time, and reaction temperature.

Line 488-490: the wild type CgASNase and the double mutant enzyme L42T/S213N were used to pre-treat the dough under the same and optimal treatment conditions.

Line 497-500: The results of the study showed that the double mutant enzyme had similar enzymatic properties to the wild type. Therefore, these findings clearly demonstrate that the mutant enzyme with enhanced thermal stability has the ability to significantly decrease the acrylamide content in biscuits.

Point 4: Overall, the English language is fine, but some points to fix are present throughout the text (e.g. line 56).

Response 4: Thanks for referee’s affirmation and suggestions. We have revised the manuscript (p. 1, line 11-12; p. 1, line 34-35; p. 2, line 62-70; p. 2, line 75-76; p. 2, line 96-97; p. 3, line 123-126; p. 3, line 133-134; p. 4, line 191-192; p. 5, line 199-200; p. 6, line 258-260; p. 8, line 348-350; p. 12, line 439-455; p. 13, line 478-484; p. 14, line 500-504).

Line 11-12: L-asparaginases could effectively regulate the formation of acrylamide at the source. However, current L-asparaginases exist drawbacks such as poor thermal stability…

Line 34-35: In modern times, baking and frying have become indispensable elements in the daily lives of humans.

Line 62-70: However, there is a scarcity of commercially approved microbial-derived L-asparaginases available for use in the food industry. Currently, only two enzymes, Acrylaway® (Novozymes A/S, Denmark) derived from Aspergillus oryzae and Pre-ventASeTM (DSM, Netherlands) derived from Aspergillus niger [15, 16], have been recognized as "GRAS" (Generally Recognised as Safe) by the United States government. Additionally, the World Health Organisation (WHO) included them as food additives in 2008 (Series 59) and 2009 (Series 60), respectively [17]. Unfortunately, the existing microbial-derived L-asparaginases suffer from limitations, such as low catalytic activity, poor thermal stability, and inadequate substrate specificity…

Line 75-76: Enzymes that possess the desired thermal stability exhibit notable advantages in terms of…

Line 96-97: Consensus design represents a more promising approach to improve enzyme performance, as it…

Line 123-126: Previous research has obtained C. glutamicum L-asparaginase (CgASNase) with high catalytic activity and substrate specificity, which effectively inhibits the formation of acrylamide in fried potato products.

Line 133-134: Finally, the optimal treatment condition of the mutant enzyme for biscuits application was determined using…

Line 191-192: The enzyme activity assay consists of a reaction system and a color development sys-tem.

Line 199-200: Similarly, a standard curve for enzyme activity was obtained using ammonium sul-phate solution as a standard: y=48.964x+0.0014 (x represents the concentration of NH4+

Line 258-260: This process was repeated by adding 10 mL of ultrapure water to the precipitate, then the supernatants were collected together.

Line 348-350: Notably, the specific activity of the double mutant enzyme L42T/S213N (1931.01±25.84 IU/mg) did not exhibit a significantly (p>0.05) decrease when compared to the mutant enzyme L42T…

Line 439-455: the acrylamide reduction in the treated biscuits was above 90% compared to those without enzyme treatment……the exchange rate of water molecules in the dough was reduced, which can lead to a decrease in the reaction rate of the enzyme. Conversely, when the temperature was excessively high, the evaporation of water in the dough resulted in an uneven distribution of water within the dough……

Line 478-484: The ANOVA analysis of the orthogonal experiments revealed that the enzymatic reaction times had no significant impact on the reduction of acrylamide. To further investigate the effect of enzymatic reaction times on acrylamide reduction, the combination A2B2C2 was included in the verification process. The results indicated that the enzymatic reaction times had no significant effect on the acrylamide content in the biscuits (p>0.05) (Table S7), which aligned with the findings of the one-factor experiment. Consequently, the optimal treatment condition was determined to be…

Line 500-504: Gazi et al. [57] investigated the effect of the enzyme dosage, dough resting times and temperatures, mixing speeds and times, and recipe components on the acrylamide reduction of thin crisp biscuits, and determined the optimal treatment condition, under which the acrylamide reduction reached 80%, with a lower inhibition than that of the mutant enzyme L42T/S213N.

Round 2

Reviewer 3 Report

Comments and Suggestions for Authors

The Authors addressed the reviewer's comments and concerns. The revised version is suitable for publication.